# Intravitreal DEX Implant for the Treatment of Diabetic Macular Edema: A Review of National Consensus

**DOI:** 10.3390/pharmaceutics15102461

**Published:** 2023-10-13

**Authors:** Roberta Spinetta, Francesco Petrillo, Michele Reibaldi, Antonia Tortori, Maria Mazzoni, Cristian Metrangolo, Francesco Gelormini, Federico Ricardi, Antonio Giordano

**Affiliations:** 1Torino Eye Hospital, Via Juvarra 19, 10024 Torino, Italy; spinetta.roberta@gmail.com; 2Department of Medical Sciences, Eye Clinic, Turin University, 10024 Turin, Italy; mreibaldi@libero.it (M.R.); francesco.gelormini@hotmail.it (F.G.); federico.ricardi@gmail.com (F.R.); 3Ophthalmology Unit, Surgery Department, Piacenza Hospital, 29121 Piacenza, Italy; antonia.tortori@gmail.com; 4University Center for Studies on Gender Medicine, University of Ferrara, 44124 Ferrara, Italy; maria.mazzoni@unife.it; 5Department of Neuroscience and Rehabilitation, University of Ferrara, 44121 Ferrara, Italy; 6Ophthalmology Unit, Ospedale di Circolo e Fondazione Macchi, ASST Sette Laghi, 21100 Varese, Italy; metrangolo.cristian@gmail.com; 7Sbarro Institute for Cancer Research and Molecular Medicine, Center for Biotechnology, College of Science and Technology, Temple University, Philadelphia, PA 19122, USA; president@shro.org

**Keywords:** diabetes, diabetic macular edema, DME, dexamethasone, DEX, intraocular implant

## Abstract

Diabetic macular edema (DME)’s therapeutic approach can frequently be challenging. The purpose of the review is to propose evidence-based recommendations on the employment of intravitreal dexamethasone implants (DEX) when approaching patients suffering from DME. Seven national consensuses redacted by different groups of retina specialists from Europe and Asia were examined and confronted. Each consensus was redacted utilizing a Delphi approach, in person meetings, or by reviewing the literature. DEX can be studied as a first-line strategy in individuals suffering from DME with inflammatory OCT biomarkers, in vitrectomized eyes, in patients with recent cardiovascular events, in pregnant women, in patients scheduled to undergo cataract surgery or with poor compliance. The other parameters considered were the indications to the DME treatment, when to switch to DEX, the definition of non-responder to anti-VEGFs agents and to the DEX implant, whether to combine DEX with laser photocoagulation, the association between glaucoma and DEX, and the management of DEX and the cataract. Although several years have passed since the introduction of DEX implants in the DME treatment, there is still not a unified agreement among retina specialists. This paper compares the approach in the DME treatment between countries from different continents and provides a broader and worldwide perspective of the topic.

## 1. Introduction

In patients with a diagnosis of diabetes, ocular complication may occur, and vision loss is mainly due to the development of diabetic retinopathy (DR) and diabetic macular edema (DME) [1]. Despite the therapeutic management of DME improving significantly in recent years, it still represents a clinical challenge as the global prevalence of diabetes is expected to rise from 415 million individuals in 2015 to reach 642 million by the year 2040 [2].

Vision loss is widely recognized for its adverse effects on a patient’s physical capabilities, constraining their capacity to engage in daily tasks. Additionally, the burden from frequent intravitreal injections has a distinct impact on a patient’s quality of life. This includes repercussions such as the necessity to take time off from work and an increased reliance on caregivers for support.

Other emotional factors, encompassing feelings of frustration, treatment-related anxiety, and needle phobia, should all be taken into account [2,3].

A European study involving 131 patients, comprising 86 with DME and 45 with retinal vascular occlusion (RVO), was conducted to investigate the influence of injection therapy on the quality of life of these individuals. This study emphasizes the necessity for many DME patients to seek medical attention not only from ophthalmologists but also from various other specialists. Over a span of six months, it was observed that more than half of the DME patients had an average of 19.1 appointments with multiple healthcare professionals, emphasizing the systemic complications associated with the condition [3].

Furthermore, a study that compared appointment patterns between patients with DME and those with neovascular macular degeneration (nAMD) revealed that patients with DME were more likely to cancel or not attend their appointments. This trend may be linked to the higher frequency of healthcare appointments required by DME patients each year [4].

The data from pivotal randomized controlled trials (RCT) demonstrated positive functional and anatomical responses and gained approval by different drug agencies of ranibizumab, aflibercept, the DEX implant (DEX-I), and the fluocinolone implant [5,6,7,8,9,10,11].

A comprehensive evaluation of the patient’s medical history and a detailed ophthalmological assessment is essential to provide the ophthalmologist with the necessary information for selecting the most suitable therapy.

Moreover, in patients with DME, it is pivotal to identify if the diabetes management is accurate as well as be aware if recent cardiovascular events occurred [7].

In 2017, Euretina proposed recommendations for the treatment in DME [8]; since then, many other national guidelines or consensus were published in order to support the clinical decisions and physician demands for updated documents in consideration of the newer evidence in regard to the available treatments. The present report aims to give an overview of the recommendations for the use of dexamethasone implants for the treatment of DME, comparing the different guidelines and consensus currently available in the literature. In this paper, only the guidelines that express a national consensus and are written in English are discussed, since the main objective was highlighting the principals points in common and the principals differences between countries. We excluded papers that did not fully address the use of the DEX implant.

The guidelines we discuss in the paper do not follow the same methodology; in fact, some are based on the Delphi method, a technique used to obtain answers to a clinical problem from a group of independent experts through several stages, while others are based on the simple agreement of a panel of experts.

## 2. Guidelines from Spanish Experts [9]

A panel of retina experts, from 11 Spanish hospitals, established consensus-based recommendations, during six meetings held from February to October 2019, for the management of patients affected by DME (Tables 1–3) [9].

According to the results of the OBTAIN study [10], the expert panel suggested observation in patients with high visual acuity (VA) (20/25 or better) and recommended the initiation of treatment when the VA is less than 20/25. Moreover, the panel focused on optical coherence tomography (OCT) biomarkers that may guide the choice of the pharmacological agent and help clinicians to predict the anatomical and/or functional response. 

Working in collaboration, the experts identified image biomarkers such as serous retinal detachment (SRD), hyperreflective dots (HRDs), disorganization of the retinal inner layers (DRIL), outer retinal layer (ORL), and central macular thickness (CMT) that should be evaluated at baseline when selecting the treatment strategy. 

First, the panel members agreed to use DEX implants rather than anti-vascular endothelial growth factor (VEGF) injections for treating DME with SRD due to the anti-inflammatory activity of DEX. Indeed, the presence of SRD has been associated with a higher concentration of inflammatory cytokines [11,12]. Although it is known that VEGF plays a role in DME etiology, it is not the only inflammatory factor involved.

Second, they all agreed that large empty cysts, seen on OCT, are usually related to more advanced chronic stages of the disease, in which the DEX implant could be considered as a first-choice therapy. Likewise, DME cases characterized by a high total volume may be a candidate for the first-line treatment with DEX being associated with a greater inflammatory component [13]. 

In addition, although the etiology of HRDs has not been clarified yet, it seems to indicate a prevalent inflammatory condition [14]. For this reason, experts agreed that the DEX implant would be the treatment of choice of DME when HRDs are detected on OCT. 

It is known that DRIL [15] and the outer retinal layer (ORL) [16] represent strong prognostic factors in DME patients. However, it could be difficult to properly evaluate the retinal layers in order to predict the evolution of the disease at baseline. In these patients, the DEX implant might facilitate an adequate assessment of the ORL status since it provides a rapid anatomic response [9]. 

Finally, the panel agreed that intravitreal anti-VEGF injections or the DEX implant could be indistinctly used as a first-choice therapy in cases associated with outer nuclear layer (ONL) damage, which is a marker of poor visual prognosis in DME. In this regard, Fonollosa and colleagues [17] found a statistically significant positive effect of the DEX implant on ONL.

## 3. Guidelines from Asian Experts [18]

The evidence-based treatment guidelines in the management of DME in an Asian population were provided by an expert panel of 12 retinal specialists, who responded to a questionnaire developed using a Delphi questionnaire as a guide (Tables 1–3) [18]. 

By reviewing the recent clinical evidence, this panel’s consensus guidelines aimed to identify cases in which intravitreal DEX should be considered as a first-line therapy and provide evidence-based management considerations for potential steroid-related complications such as intraocular pressure (IOP) elevation and cataract progression. Consensus on any recommendation was achieved when 9 of the 12 Asian panel members (75%) were in agreement.

This panel recommended the use of the DEX implant as the first choice in patients with high-risk cardiovascular disease, poor compliance, central macular thickness (CMT) > 500 µm, a history of vitrectomy, as well as in patients scheduled to undergo cataract surgery or in pseudophakia.

As shown in the study by Rezkallah and colleagues [19], vitrectomy has no impact on the efficacy and safety profile of DEX implants for DME. Conversely, a decrease in efficacy of anti-VEGF agents has been demonstrated in vitrectomized eyes [20].

Regarding the utility of performing fluorescein angiography before setting the treatment plans, panel members were unable to reach a consensus due to the conflicting evidence on the role peripheral ischemia plays in persistent macular edema [18]. 

Steroids are useful in DME patients with a high inflammatory component because they have a more significant anti-inflammatory effect compared with anti-VEGF agents. However, appropriate counseling for possible side effects should be provided to the patients undergoing the intravitreal steroid treatment. Patients with stable glaucoma treated with monotherapy can undergo DEX implantation after a careful risk–benefit evaluation. Choosing the most effective treatment option takes precedence over the risk of cataract development or progression, as vision can be restored via cataract surgery [18].

Routine IOP check can be performed at 6 weeks post-DEX implantation in patients with no other ocular comorbidities. Instead, glaucoma patients should be assessed at week 1, week 2, and 4 to 6 weeks post-treatment to identify increased IOP and promptly modify the treatment. 

If the response to the DEX implant is adequate (VA greater than or equal to 6/12 and CMT < 300 µm, or improvement in CMT > 10%), retreatment may be considered on a 4 to 6 monthly basis. Otherwise, the Asian panel recommended to switch to anti-VEGF agents.

Overall, consensus guidelines for the management of DME in Asian populations supported the use of intravitreal DEX not just as a second-line choice but also as a first-line therapy in pseudophakic, vitrectomized eyes, and in patients undergoing cataract surgery or with poor compliance. Finally, Asian clinical guidelines addressed the precautions to be taken during the intravitreal DEX treatment due to its possible side effects on IOP, glaucoma, and cataract progression [18].

## 4. Guidelines from French Experts [21]

In Europe, the use of anti-VEGF agents (Ranibizumab-Lucentis, Aflibercept-Eylea, brolucizumab-Beovu, and Faricimab-Vabysmo) and the biodegradable dexamethasone implant (Ozurdex) have been approved for the treatment of diabetic macular edema [22]. However, the extensive use of anti-VEGF as the first line of treatment has shown a high percentage of patients, between 25 and 40%, who did not respond optimally to therapy. Furthermore, the presence of some comorbidities and the need for frequent injections made thetreatment with anti-VEGF even more difficult. The biodegradable DEX implant represents an important alternative to the use of anti-VEGFs [23]. 

In France, one of the countries with the most experience in the use of dexamethasone implants, a national consensus was developed from April 2020 to September 2020 to establish guidelines for the use of dexamethasone implants via the DELPHI method (Tables 1–3) [21]. In particular, the opinions of 39 retinologists were collected on the basis of a questionnaire written by a Steering Committee made up of six highly experienced ophthalmologists. 

The participants expressed their opinion through their agreement with the chosen questions using a nine-point Likert scale, where 1 was considered “strongly disagree” and 9 “strongly agree”. For a single question, a strong consensus was considered when more than 75% of the scores were ≥7 and the median score was ≥8. If only one of these two parameters was satisfied, the statement was considered to have obtained a good consensus. Finally, failure to achieve at least one of the two parameters was considered as a “No consensus agreement” [24,25]. The questionnaire took into consideration five key aspects of the management of diabetic macular edema with DEX implantation: pathophysiology, indications for the dexamethasone implant as a first-line treatment, time to retreatment, efficacy criteria, and safety. 

Regarding pathophysiology, experts have recognized, with strong consensus, the importance of inflammation in the development of diabetic macular edema and its presence in all phases of the disease. Moreover, they all agree with “strong consensus” about the anti-inflammatory, anti-angiogenic, and blood–retinal barrier stabilizing activity of Ozurdex. However, “no consensus agreement” was reached regarding the ability of OCT to identify the inflammatory biomarkers and guide the treatment choice. 

Concerning the indications for dexamethasone implantation, as a first-line treatment, a strong consensus was reached in the case of patients who have had a cardiovascular event in the past 3 months, patients with poor compliance, and vitrectomized patients. Furthermore, a good consensus was reached in the use of Ozurdex as a possible first choice even in the case of patients with non-proliferative diabetic retinopathy for whom cataract surgery was planned and in patients who showed on OCT the presence of the serous detachment of the neuroepithelium, hyperreflective foci, and numerous hard exudates.

Regarding the frequency of the Ozurdex treatment, retinologists strongly agreed that the mean time interval between two injections was 3–5 months and that it was not necessary to wait for a reduction in visual acuity to retreat the patient with DEX-I. However, no consensus has been reached on whether patients with dry retina can be treated with a proactive regimen (as in the case of anti-VEGF [26,27]). 

Regarding the criteria to evaluate the efficacy of DME therapy, the panelists agreed in defining the functional non-response as visual acuity improvement < 5 ETDRS letters and the anatomical non-response as a reduction in the central retinal thickness < 20% and/or the lack of significant intraretinal cyst improvement. Furthermore, a good consensus was reached that it was necessary to wait until after two injections of DEX without anatomical/functional response to consider it not effective. 

In regard to the safety of the dexamethasone implant, the retinologists took into consideration the two aspects that are mainly of concern in the use of Ozurdex: the increase in intraocular pressure, and the development of cataracts. For the increase in IOP, the panelists agreed that in most cases in which intraocular pressure increases, topical hypotensive drugs are sufficient for its control and that an IOP control is necessary between the first and second month post-DEX-I. 

The experts agreed that the risk of DEX-I-induced IOP is less significant in DME compared to uveitis and retinal vein occlusion, and that the risk of dexamethasone-induced glaucoma is inferior compared to triamcinolone. Furthermore, a topical steroid loading test was not considered necessary prior to dexamethasone implant use, and the presence of effective filtering surgery was not considered a contraindication to DEX-I use. 

Finally, there is a known risk of cataract surgery after DEX-I in patients who already have a diagnosis of cataracts [8]; however, 63% of French retinal experts estimated that the risk of developing a cataract after two injections of dexamethasone on a clear lens was low (therefore without reaching a good consensus).

The panelists agreed to recommend an injection of DEX, about 2 weeks before or at the time of cataract extraction, to reduce post-operative edema.

## 5. Guidelines from the Emirates Experts [28]

In the second half of 2019, a group of expert retinologists from the Emirates Society of Ophthalmology provided guidelines for the optimal management of diabetic macular edema. The main aspects on which the authors focused were the initiation of therapy, systemic management of patients with DME, and treatment recommendations (Tables 1–3) [28]. 

Regarding the initiation of therapy, the panel members stressed that the goal of therapy is to improve vision whether it diminished or to stabilize it, and finally, to prevent structural damage to the macula [29]. In this context, the conditions that require a treatment are best-corrected visual acuity (BCVA) lower than 20/30, and/or the presence or sign of diabetic macular edema (DME) on optical coherence tomography (OCT) with a central retinal thickness (CRT) of at least 300 um or patients with BCVA better than 20/25 but symptomatic for visual disturbances due to DME, and/ or CRT less than 300 um with OCT features consistent with center-involving macular edema [30,31]. 

As far as the systemic management of patients is concerned, the treatment of diabetic macular edema is effective regardless of glycated hemoglobin; thus, it is essential not to postpone therapy while waiting for systemic compensation [32,33]. However, a good control of diabetes, glycated hemoglobin, hypertension, dyslipidemia, renal insufficiency, and sleep apnea allows for a significant improvement in the management of macular edema [34,35]. 

The recommendations for the treatment of macular edema were finally divided into two groups: the treatment of non-center-involving DME and the treatment of center-involving DME. 

Non-center-involving DME is defined as macular thickening not involving the central subfield zone (1 mm in diameter on OCT). In this case, monitoring is essential until the progression toward the center is evidenced, or in selected cases, anti-VEGF or photocoagulation laser therapy may be recommended [18,36].

Instead, the center-involving DME is defined as the macular thickening involving the central subfield zone on OCT. In this case, anti-VEGFs represent the first line of treatment while the dexamethasone implant represents a valid alternative in the numerous situations in which anti-VEGFs are contraindicated or ineffective [37,38]. 

In particular, anti-VEGF drugs require a loading dose of 3–6 monthly injections followed by a therapeutic regimen based on the patient’s response. The estimated number of optimal anti-VEGF injections in the first year of treatment is 8–9 [39,40]. A poor response to the anti-VEGF treatment after the loading dose is defined as a failure to gain at least five letters of vision and/or reduce CRT by 10% [41]. 

On the other hand, the biodegradable dexamethasone implant has a peak of action between six and eight weeks, and its effectiveness generally lasts about 4 months. For this reason, the estimated number of dexamethasone implants in the first year of treatment is 3–4 [42]. Approximately, 10% of patients treated with DEX-I have an increase in IOP > 25 mmHg, and there is also an increased risk of developing cataracts [43,44]. The DEX implant can be an option in the case of contraindications to the use of anti-VEGF (as in the first 3 months after a heart attack/stroke), or poor compliance with the treatment regimen of anti-VEGF, or poor response to the anti-VEGF treatment after the loading dose. During pregnancy, the first-line therapy should be glucose control and laser photocoagulation. However, in severe cases of DME, the DEX implant could be an option, but it has to be offered only in the second/third trimesters and carefully discussed with the patient [41,45,46,47]. Moreover, it can also be recommended in vitrectomized, pseudophakic, or chronic DME patients. Furthermore, the experts agreed that the use of dexamethasone implants in patients with submacular fluid, hyperreflective foci, intra-retinal cysts, and a disorganization of the continuous inner retinal layer (also referred to as DRIL) may give better results than in patients without these features [48]. Finally, the panel members identified contraindications to the use of DEX implants such as the presence of active or suspected ocular or periocular infection, advanced or non-compensated glaucoma (requiring more than three medications), interruption of the posterior capsule (YAG capsulotomy excluded), eyes with aphakia, and hypersensitivity to the dexamethasone implant [49].

## 6. Guidelines from Italian Experts [50]

The DEX implant role in the management of DME is still not defined. For this reason, this survey adopting a Delphi-based approach aimed to provide some recommendations that are useful for ophthalmologists treating DME when choosing DEX in daily practice.

A Steering Committee of four Italian medical retina specialists after reviewing the literature formulated 30 relevant topics of discussion. A group of 40 retinal specialists from across Italy answered the questionnaire. The 30 relevant statements focused on the use of DEX in DME and mainly deals with these six areas: (1) etiopathogenesis of DME; (2) first-line treatment with DEX; (3) safety of therapies for DME; (4) switch to DEX in previously antiVEGF-treated patients; (5) DME proliferative DR; and (6) burden of treatment. When ≥75% of the panelists “very much agree” or “agree” with a particular statement of the questionnaire, consensus for that particular statement was reached (Tables 1–3).

According to the different topics, the following consensus were reached:(1)Consensus on the etiopathogenesis of DME.

Broad consensus among the experts regarding the etiopathogenesis of DR and DME as a multifactorial complication of diabetes where inflammation plays a determining role [48,51,52,53].

(2)Consensus on the use of the DEX implant as a first-line treatment for DME.

Firstly, a strong agreement on the efficacy of the steroid therapy in modulating the major pathogenic aspects of DME resulted [14,54,55].

Secondly, the experts agreed that the DEX implant is a valid alternative in the first-line therapy of DME due to its good efficacy, in particular for pseudophakic patients, and that it should be the first-choice therapy in vitrectomized patients [56,57,58,59]. Moreover they agreed that DEX is the therapy option to choose for patients who have had an arterial thromboembolic event (ATE) in the past 4 months due to the lack of an evident relation between cardiovascular events and the DEX implant.

Finally, regarding the retreatment timing for DEX, the panelists held that a Pro Re Nata (PRN) regimen has to be chosen as more suitable and effective than the fixed regimen with a 6-month wait proposed when DEX was first introduced into ophthalmology practice.

(3)Consensus on the safety of therapies for DME.

The panel agreed on the importance of evaluating the cardiovascular safety profile in diabetic patients before choosing the intravitreal treatment with an anti-VEGF; moreover, it agreed in considering the DEX implant among the steroids as the one with the best ocular tolerability [60]. It is indeed reported that two-thirds of the patients do not need antiglaucoma drops and that 59.2% of treated patients would need phacoemulsification after the DEX implant [61].

Almost all panelists agreed on using the DEX implant in patients in topical antiglaucoma therapy. In the MEAD study, it highlights the non-cumulative effect on the patient’s intraocular pressure of more DEX implants; that the possible rise of IOP generally lasts a short time and is easily treated with topical therapy; and finally, the MEAD study underlines that the IOP rises are less frequent and less severe in DEX patients than in those treated with intravitreal fluocinolone acetonide or triamcinolone acetonide [61].

Moreover, it was reckoned that the topical steroid-loading test is not useful nor required before the DEX implant injection.

(4)Consensus on the use of DEX DDS in already antiVEGF-treated patients:

The panel agreed on considering a patient as a clinical antiVEGF non-responder according to both the CRT and VA evaluation and that the treatment is not effective if after six monthly antiVEGF injections, there is no improvement in the OCT morphology, as suggested in some pivotal trials such as Protocol T [62]. On the other hand, the experts did not agree in considering a patient as a clinical non-responder if there is still edema after three injections of ranibizumab following the evidence of the DRCR net study where gains in BCVA were seen despite a macular edema still present after several injections of ranibizumab [40].

Finally, the panel agreed on switching to the DEX implant in patients with persistent edema after the antiVEGF loading phase. In these patients, the DEX implant showed significant results both in the visual and anatomic goals [63];

(5)Consensus on the use of DEX in patients with both DME and proliferative DR:

In patients with DME and proliferative diabetic retinopathy, both the DEX implant and anti-VEGF together with pan retinal photocoagulation are recommended by the experts. Indeed, a good restoration of outer retinal layers morphology after the combined treatment of the DEX implant and panretinal photocoagulation (PRP)was shown [64].

(6)Consensus on the burden of treatment for DME using DEX:

The panel agreed on the improvement in the quality of life of patients and their families due to the lesser number of injections and control visits associated with the DEX treatment, so the experts largely agreed that the DEX treatment should be considered the treatment of choice rather than antiVEGFs in patients who have poor compliance with several medical appointments [65]. 

Secondly, the panel largely agreed that the treatment regimen of DEX is more suitable than anti-VEGF’s one in daily practice. The DME treatment protocol with antiVEGFs considered monthly injections until the best visual function possible was achieved, whereas it is reported that with DEX DDS, approximately only 4–5 injections are needed in the three years of follow up [66]. Moreover, the average number of injections in a year has been seen to be three-fold higher with ranibizumab than DEX [56,66].

In conclusion, the DEX treatment can be applied easier than antiVEGF’s one and is less demanding in terms of the health system, economic, and physical resources.

## 7. Guidelines from Saudi Arabia Experts [67]

Diabetic retinopathy and diabetic macular edema’s prevalence in Saudi Arabia’s diabetic population is respectively 19.7% and 5.7%. Patients suffering from diabetic macular edema in Saudi Arabia are often unrecorded, which may lead to the misconception of a lower incidence of DME in the kingdom compared to the world scenario [68].

The Saudi Retina Group at the end of 2020 decided to develop a consensus for the clinical diagnosis and for the therapeutic strategies of DME following the guidelines based on the most recent evidence-based medical practice (Table 1, Table 2 and Table 3) [69]. Eight DME experts, from seven different Saudi institutions, both from the public and the private healthcare system, participated in the consensus development.

The experts’ panel agreed that early treatment in patients suffering from DME is necessary in order to attain a better outcome and that the location of the edema is crucially important. 

Patients who suffer from DME localized outside of the macular center are addressed to focal laser photocoagulation as the first-line therapeutic strategy, in addition to strict monitoring of the cardiovascular risk factors, such as hypertension, smoking, and hyperglycemia [70].

Patients who suffer from DME localized in the macular center are addressed to 3 months of ranibizumab or aflibercept regimen as the first-choice treatment, regardless of if the patient previously underwent cataract surgery.

The positive response of the anti-VEGF treatment is characterized by VA improvement > 5 letters or a reduction of more than 20% in the central retinal thickness compared to the last OCT scan before treatment [10]. In patients classified as responders, anti-VEGF injections should be proceeded following the treat and extend strategy (TE): TE that has been proven to be superior in regard to better visual results despite fewer treatments and reduces the number of necessary specialist visits, being more affordable for both the hospital and the patients [71]. 

OCT examinations should be performed at regular intervals of 3 months after the anti-VEGF loading dose.

The panelists concurred with the definition to non-responders to anti-VEGF: the sign of severe macular edema, a reduction inferior to 20% in the central retinal thickness, and no improvement in terms of VA (cut-off value of 5 letters) after the loading course [72]. According to these criteria, 20–30% of patients in Saudi Arabia are classified as non-responders to the anti-VEGF treatment.

The experts stipulated that, in the case of a non-response after the first 3 injections, an early switch should be considered either to another anti-VEGF agent (in the case of minimal response to the previous injection) or to steroid injection.

Steroids implants have already been demonstrated to be efficacious in the DME treatment. In Saudi Arabia, the two steroid on-label options are the dexamethasone intravitreal implant and fluocinolone acetonide. Triamcinolone acetonide injections are proven to be effective in the DME treatment, but given their higher risk of complication (44% of patients developing glaucoma and 54% needing cataract surgery) and their off-label indication, they were not considered in the consensus drafting [73].

Steroid injections are beneficial to non-responder anti-VEGF patients, patients who previously underwent vitrectomy surgery, patients unable to be treated on a monthly basis, and cases diagnosed with recent cardiovascular or cerebrovascular events (within the last 3 to 6 months). Patients detected with acute cardiovascular comorbidities in the last 3 to 6 months should be treated with steroids as the first choice. Vitrectomized eyes could use steroid implants as the initial treatment or following 3 injections of anti-VEGF in non-responding patients.

The panelists asserted that the DEX implant should be considered also in pregnant women: mild DME developed during gestation frequently recovers spontaneously post-partum and can only be observed, whilst significant DME detected before pregnancy can be treated with the DEX implant [67].

The first choice of steroid injection should be of the dexamethasone intravitreal implant (DEX implant), whilst fluocinolone acetonide should be studied as a second alternative in the cases of patients with an unsuccessful response to a previous steroid injection. 

Finally, the panelists recommend to replace the “tractional macular edema” definition with the vitreomacular traction (VMT) definition in order to focus on the tractional force operated by the vitreous humor after an incomplete posterior detachment. These patients should undergo one trial of intravitreal anti-VEGF injection, and if they are realized as non-responders, vitreoretinal surgery should be performed.

## 8. Guidelines from Italian Experts for Patient Undergoing Cataract Surgery [74]

Following the MEAD study results, DEX implants in patients with DME showed a protective effect following cataract surgery [68]. When diabetic patients come to cataract surgery, we have to consider two important points: first, the presence of DR and/or DME may affect the functional outcome [69], and secondly, DR and DME prevalence are significant [75]. As a result, the practical guidelines regarding the management of patients with DR undergoing cataract surgery and the use of DEX implants in these cases could be useful.

Few literature deals with that topic; this paper, a synthesis of the consensus results of a panel of Italian experts, tried to cover this issue and provide a guide on the use of DEX implants in this subgroup of patients. 

Eight consensus statements resulted from the survey.

The eight consensus statements developed referred to three major areas considered: (1) rationale for the use of the DEX implant in patients affected by DME at the time of surgery; (2) patient diagnosis and selection before treatment; (3) management of patients treated with DEX.

(1)Consensus statements referred to the rationale for the use of the DEX implant in patients affected by DME at the time of surgery were:

Diabetic patients have a higher risk of developing macular edema after surgery than those who are not. It is well known that phacoemulsification increases inflammatory cytokines [76]; even without DR, diabetic patients show a relative risk of developing DME after surgery of 1.80 and 6.23 if they already have DR of some degree [70].

The more severe the grading of the DR, the higher the risk of worsening the DME following surgery, as showed by Chu et al. [70].

If there is already a DME, the risk of edema increasing post-surgery is high. Cataract surgery in patients with diabetes is followed by a DME progression in 23% to 57% of cases [71] as a result of the post-surgical inflammatory cascade surgery; that is why corticosteroids playing the role of antinflammatory agents may be useful in reducing this risk.

The increase in macular edema in the diabetic patient is more likely in the first two months after phaco.

(2)Consensus statements referring to the patient diagnosis of DME and the selection for treatment were:

An optical coherence tomography examination prior to phaco is mandatory to assess the retinal status. Patients with non-central or central DME prior to cataract surgery showed an increasing risk for the development of central-involved or more severe DME at 16 weeks post-op [72].

A diabetic patient who already shows some kind of DME and who is planned for cataract surgery should be treated. A DEX implant injection has the potential to avoid the increase in the central retinal thickness seen after cataract surgery in diabetic patients (as it reduces the inflammatory cascade), and therefore, the occurrence or the worsening of DME after surgery wards off a loss of visual acuity [73].

(3)Consensus statements referring to the optimal management were:

DEX-DDS is the treatment of choice in patients with DME who are naïve to the treatment planning for phaco. In a pool analysis of 1048 patients, the CRT increase was not recorded after phaco surgery in patients showing DME at preoperative OCT treated with DEX [66]. The use of DEX has a relatively broad timeframe, where as anti-VEGF agents should not be injected within 28 days before or after ocular surgery [72]. On this basis, and in tune with the literature, this paper suggests the DEX implant as the appropriate therapy in DME patients undergoing phaco [73].

The panel’s indication for the DEX implant is for sure a central DME, where the use of DEX to prevent DME in patients with a relatively high risk was under debate but did not reach a consensus.

DEX DDS can be implanted in the timeframe between one month before and after surgery (including during the surgery) in DME patients who must undergo cataract surgery; it improves the morphological and functional outcomes. The precise optimal timing of the DEX implant is still under debate; however, the interval should be one month before or after the surgery.

## 9. Discussion

Although several years have passed since the first DEX implant, there is no total agreement among the various guidelines. Below, we will try to analyze the key points of the use of the DEX implant in different consensuses.

### 9.1. DEX as a First-Line Treatment

Most guidelines agree in using the DEX implant as the first choice in patients with previous cardiovascular events, in the presence of poor compliance, or in vitrectomized eyes. There is also a large consensus to using the DEX implant as the first choice in pseudophakic patients, although not all guidelines agree on this point either. Some guidelines begin to consider OCT biomarkers for the use of the DEX implant as a first line. In particular, the presence of SRD and HRDs are referred to as being the most important OCT biomarkers in the choice of the DEX implant as a first line, although some authors also consider the presence of hard exudates, DRIL, and large cysts.

### 9.2. Indications to DME Treatment

An indication to the DME treatment is often not considered and represents one of those aspects where there is not yet a total agreement. In fact, as a main indication to starting the DME treatment, some guidelines consider the patient’s visual acuity while others take into account both the OCT macular thickness and visual acuity.

### 9.3. Switch to DEX Treatment

Some consensuses provides some suggestions on when to switch to the DEX implant. It is generally recommended to switch to a DEX implant when there is not a good response after the antiVEGF loading dose. 

### 9.4. DEX Implant Retreatment Protocol and Timing

Furthermore, the retreatment time of the DEX implant is sometimes considered, and it is generally suggested to repeat it in a time interval from 3 to 6 months with a good agreement among the various guidelines.

### 9.5. Definition of Non-Responder to Anti-VEGFs Agents

There is no agreement on the definition of non-responder anti-VEGFs agents, which will have to be better determined in future guidelines. Indeed, some guidelines merely indicate the estimated number in the general population of patients who do not respond to anti-VEGF while others consider the lack of improvement in OCT morphology as sometimes being associated with a failure in visual acuity improvement to report a lack of response to the anti-VEGF agents.

### 9.6. Definition of Non-Responder to DEX

Also, in this case, there is no common concordance among guidelines, although a non-improvement on visual acuity and of the retinal thickness are considered good parameters to define a non-response to the DEX implant.

### 9.7. DEX Implant in Glaucoma Patients

However, with regard to patients affected by open-angle glaucoma, most of the guidelines state that the DEX implant could be considered as a good option in stable glaucoma, while it is strongly contraindicated when intraocular pressure is not well controlled.

### 9.8. IOP Monitoring

IOP monitoring is another crucial point after the DEX implant, where it is suggested to be checked at 6 weeks post-injection and to be evaluated more frequently in the first two months in glaucoma patients.

### 9.9. Topical Steroid Loading Test Utility

The topical steroid loading test, which is a test that uses topical steroid drops in the days before injection to predict an increase in eye pressure before the DEX implant, is generally not considered as a useful way to foretell an increase in the ocular pressure.

### 9.10. DEX Implant and Cataract Development

The risk to develop cataracts is considered low after two DEX implants and it could also be considered a good option in phakic patients if it is supposed to be the most effective treatment in those patients. 

### 9.11. DME and Cataract Surgery

In the presence of DME and cataract surgery, it is suggested to use the DEX implant two weeks before cataract extraction.

In recent years, we have seen an increase in the use of the DEX implant as a first choice in DME. This may be due to a rising use of this treatment which, over the years, has shown not only good efficacy, but also a good safety profile. However, further efforts will have to be made to unify the guidelines which still have some inconsistencies. For this reason, a meeting of world experts could be useful to create shared guidelines.

## Figures and Tables

**Table 1 pharmaceutics-15-02461-t001:** Summary of the Aims, Methods, and Agreement between national consensuses.

	Spanish Consensus	French Consensus	Asian Consensus	Emirates Society of Ophthalmology Consensus	Italian Consensus	Saudi Arabia Consensus	DEX-CAT Italian Consensus
Year	2019	2020	2019	2019	2017	2020	2018
Aims	Treatment guidelines and consensus for the management of DME.	Treatment guidelines and consensus for the management of DME with DEX-I.	Treatment guidelines and consensus for the management of DME.	Treatment guidelines and consensus for the management of DME.	Treatment guidelines and consensus for the management of DME with DEX-I.	Treatment guidelines and consensus for the management of DME.	Guidelines for dexamethasone implants in patients with diabetic macular edema undergoing cataract surgery.
Methods	A group of expert retinologists developed a consensus related to clinical management of patients with DME. In the first meeting, the panel selected and agreed first to a list of topics related to the clinical management of DME patients Attending to these subjects, the experts developed a list of questions. These questions were discussed, updated literature was reviewed, and responses were agreed in different meetings held from February to October 2019 (six meetings in total).	Modified Delphi consensus: The expert opinion gathering took place in two voting rounds based on a questionnaire drawn up by a steering committee (SC). The participants indicated their level of agreement using a nine-point Likert scale ranging from 1 (strongly disagree) to 9 (strongly agree).	Delphi questionnaire: A total of 47 questions were developed. An expert panel of 12 retinal specialists responded to the questionnaire at 2 separate occasions in a masked fashion.	Based on evidence taken from the literature and published trials of therapies, as well as the consensus opinion of a representative expert panel convened by the Emirates Society of Ophthalmology. Two in-person meetings in Dubai in June 2019 and December 2019.	Delphi consensus: The Steering Committee was composed of four experts who oversaw the preliminary review of the literature and the formulation of 30 relevant statements A panel consisting of 40 ophthalmologists/retinal specialists from across Italy.	Eight consultant ophthalmologists participated in the consensus development and represented 7 Saudi specialized institutions: 6 from the government sector and one from the private sector.	A survey composed of 10 questions regarding the use of DEX-DDS in patients with DME undergoing cataract surgery was drafted by two panelists. The results of the survey were then discussed at the meeting. Consensus was considered to be reached when all experts agreed on a statement; if a full agreement was not reached, the statement was used as the basis for discussion.
Consensus Agreement	Not applicable	Strong consensus was reached for a statement when more than 75% of the scores were ≥7 and the median score was ≥8. When only one of these two parameters was satisfied, the statement was considered to have obtained a good consensus.	Consensus on any recommendation was considered “achieved” when 9 of the 12 panelists (75%) were in agreement.	Not applicable	Voting results were scored as “very much agree”, “agree”, “neither agree nor disagree”, “disagree”, and “very much disagree.” Consensus was considered when ≥75% of respondents voted “very much agree” or “agree” with a particular statement.	Not applicable	Not applicable

**Table 2 pharmaceutics-15-02461-t002:** Indications for DEX treatment.

	Spanish Consensus 2019	French Consensus	Asian Consensus	Emirates Society of Ophthalmology Consensus	Italian Consensus	Saudi Arabia Consensus
DEX as a first-line treatment	DEX implant should be considered the treatment of choice rather than antiVEGFs if SRD *, large empty cysts, high total volume, and HRDs ** are seen.	Cardiovascular event, poor compliance, vitrectomized patients, patients with RDNP with cataract surgery planned. First-choice treatment when SRD or HRDs or hard exudates are present.	In patients with high-risk cardiovascular disease, poor compliance, CMT *** > 500 micron, vitrectomized, pseudophakic, or scheduled to undergo cataract surgery.	In vitrectomized or pseudophakic patients, in pregnant woman, in the 3 first months after a heart attack/stroke, during breastfeeding, in case of poor compliance with anti-VEGF regimen or poor response to anti-VEGF treatment after loading dose, in case with chronic DME. Patients with SRD, HRDs, DRIL ****, intra-retinal cysts may respond better to DEX implant.	An alternative in first-line therapy for pseudophakic patients; it should be the first choice in vitrectomized patients. It is the most appropriate treatment in patients with a history of ATE (arterial thromboembolic event) in the past four months.	In patients with CV events in the last 3 to 6 months, in vitrectomized eye (at first or following the loading dose of antiVEGF in non-responding patients), in pregnant women.
Indications to DME treatment	Observation if VA is 20/25 or better; treatment if VA is <20/25.	Not considered	Not considered	BCVA ≤20/30, CMT *** ≥300 micron or symptomatic patients with VA> 20/25 and/or CMT <300 micron with OCT features consistent with center-involving macular edema.	Not considered	In presence of a clinically significant DME according to ETDRS definition.
Switch to DEX treatment	Not considered	Not considered	Not considered	Not considered	In patients with persistent edema after the loading-phase antiVEGF treatment.	In case of a non-response after an antiVEGF loading dose, a switching to an another antiVEGF or to steroid injection.
DEX implant retreatment protocol and timing	Not considered	3–5 months.	If a good response is observed, retreatment on a 4 to 6 monthly basis.	Not considered	Pro Re Nata regimen. A fixed regimen involving a 6-month wait is not appropriate.	Not considered
Definition of non-responder to anti-VEGFs agents	Not considered	Estimated between 25 and 40% percent of patients.	Not considered	Not considered	If after six monthly injections, there is no improvement in morphology by OCT.	VA improvement < 5 letters or CMT improvement < 20% or sign of massive edema after the initial loading dose.
Definition of non-responder to DEX	Not considered	After 2 consecutive injections: visual acuity improvement< 5 ETDRS letters and central retinal thickness reduction <20% and/or the absence of cysts improvement.	VA lower than 6/12 and CMT *** > 300 micron or improvement in CMT *** < 10%.	Not considered	Not considered	Not considered

* SRD: serous retinal detachment; ** HRD: hyperreflective dots; *** CMT: central macular thickness; **** DRIL: disorganization of retinal inner layers.

**Table 3 pharmaceutics-15-02461-t003:** DEX implantation in patients with glaucoma or cataracts.

	Spanish Consensus 2019	French Consensus	Asian Consensus	Emirates Society of Ophthalmology Consensus	Italian Consensus	Saudi Arabia Consensus
DEX implant in glaucoma patients	Not considered	Patient with stable glaucoma.	Patients with stable glaucoma treated with monotherapy can undergo DEX implant.	DEX implant should not be used in patients with advanced glaucoma requiring more than three medications.	Consensus not reached regarding the use of DEX implant in patients with ocular hypertension controlled via topical therapy.	Not considered
IOP monitoring	Not considered	Monitoring IOP between the first and second month after DEX implant.	Routine IOP check at 6 weeks post-DEX implant; in glaucoma patients at week 1, week 2, and 4 to 6 weeks post-treatment.	Not considered	Not considered	Not considered
Topical steroid loading test utility	Not considered	Not needed	Not considered	Not considered	Not useful	Not considered
DEX implant and cataract development	Not considered	Low risk to develop cataracts after 2 injections.	DEX implant also in phakic patient if it is supposed to be the most effective treatment in that patient.	Not considered	Not considered	Not considered
DME and cataract surgery	Not considered	DEX implant 2 weeks before cataract extraction.	Not considered	Not considered	Not considered	Not considered

## Data Availability

Not applicable.

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
