# Peer review of "Intravitreal DEX Implant for the Treatment of Diabetic Macular Edema: A Review of National Consensus"

_pharmaceutics, 2023, doi:10.3390/pharmaceutics15102461_

Round 1
Reviewer 1 Report
This is a great paper and I enjoyed reviewing it. It is the kind of paper that helps clinicians in practice to make decisions easier.
The table at the end of the paper is really useful as it summarises the findings in an easy to follow way.
Below are some minor queries but I think Q1 and Q2 need to be addressed adequately
1. What was the inclusion criteria for the guidelines you included and was there an exclusion criteria
2. In the introduction clarify that the data included did not all follow the same methodology e.g. Delphi or cover the same number of questions.
3. Some abbreviations used without prior full names e.g. RPND on line and fc laser on line or DEX DDS;
4. On line 317 Finally the panelists held that a Pro Re Nata (PRN) regimen… you do not say whether this was consensus or just agreement
5. Language used in line 344-345 needs editing for better clarity to the reader
6. You use terms like strong consensus, good consensus, poor consensus It is not clear where on the spectrum form strong to poor BROAD falls. Broad is used in line 357 for example. Perhaps in the introduction you could clarify that consensus or agreement is described differently in the included data
7. There is need for consistency with the subheadings which range from Guidelines from Saudi Arabia to Guidelines form Italian expert to Guidelines form to Guidelines from the Emirates Society of Ophthalmology…. Why are they not all experts? When you use the term expert does it mean the guidelines came form one person. e.g. French expert on line 169 and Italian expert on line 287
8. The location of the edema crucially important. Line389 A word is missing
9. Perhaps you could add a main reference for the main publication that you are quoting at the subheading level
10. There is no clear description of how consensus was reached by the Saudi team
11. Clarify line 423: “DME during pregnancy pass spontaneously post-partum and doesn’t require implants “
12. Please add a reference for this statement on line 489 “anti-VEGF agents should not be injected within 28 days before or after ocular interventions.”
The list of edits above also points out areas where the English needs to be modified for clarity
Author Response
This is a great paper and I enjoyed reviewing it. It is the kind of paper that helps clinicians in practice to make decisions easier.The table at the end of the paper is really useful as it summarises the findings in an easy to follow way.Below are some minor queries but I think Q1 and Q2 need to be addressed adequately.
We thank the reviewer for the comments.
- A: What was the inclusion criteria for the guidelines you included and was there an exclusion criteria
R: We thank the reviewer for the comment. We included only the papers published after the Euretina Guidelines in 2017 and written in English. Furthermore we selected only the guidelines that express a national consensus, since the main objective of our report was to compare the indication of the different countries. We excluded papers that did not fully address the use of DEX-implant.
We have added this explanation in the introduction, Line
- A: In the introduction clarify that the data included did not all follow the same methodology e.g. Delphi or cover the same number of questions.
R: We thank the reviewer for the comment. We modified the introduction as requested, Line
- A: Some abbreviations used without prior full names e.g. RPND on line and fc laser on line or DEX DDS;
R: We thank the referee for the comment. We carefully control all the abbreviations.
- A: On line 317Finally the panelists held that a Pro Re Nata (PRN) regimen… you do not say whether this was consensus or just agreement.
R: We thank the reviewer for the comment. As we have explained in the text, in this subchapter are mentioned only the findidngs with consensus between expert.
- A: Language used in line 344-345 needs editing for better clarity to the reader
R: We thank the referee for the comment. We rewrite the sentences.
- A: You use terms like strong consensus, good consensus, poor consensus It is not clear where on the spectrum form strong to poor BROAD falls. Broad is used in line 357 for example. Perhaps in the introduction you could clarify that consensus or agreement is described differently in the included data.
R: We thank the reviewer for the comment. We have explained that the methods between papers were different. For Italian expert consensus Broad consensus means that a considerable higher than 75% percentage of panelists agreed with a specific query.
- A: There is need for consistency with the subheadings which range fromGuidelines from Saudi Arabia to Guidelines form Italian expert to Guidelines form to Guidelines from the Emirates Society of Ophthalmology…. Why are they not all experts? When you use the term expert does it mean the guidelines came form one person. e.g. French expert on line 169 and Italian expert on line 287.
R: We thank the reviewer for the comment. We Have modified the subheadings following the suggestion.
- A: The location of the edema crucially important. Line389 A word is missing
R: We thank the reviewer for the comment. We modified as requested
- A: Perhaps you could add a main reference for the main publication that you are quoting at the subheading level
R: We thank the reviewer for the comment. We have added the references as requested.
- A: There is no clear description of how consensus was reached by the Saudi team
R: We thank the reviewer for the comment. In the guidelines of Saudi arabia experts it is not explained how the consensus was reached. We have Explained in the introduction that the methods between the papers were different.
- A: Clarify line 423: “DME during pregnancy pass spontaneously post-partum and doesn’t require implants “
R: We thank the reviewer for the comment. We agree and we have clarified the information in the text.
- A: Please add a reference for this statement on line 489 “anti-VEGF agents should not be injected within 28 days before or after ocular interventions.”
R: We thank reviewer for the comment. We modified the text as requested
Reviewer 2 Report
Dr Spinetta et al provides a review of several national recommendations on the management of DME focusing on the use of biodegradable intraocular corticosteroid implants, mostly dexamethasone. Although, this analysis sounds interesting, I find data from different sources not equal at least in authors’ representation. It is not clear for me why Italian guidelines for cataract are put in a separate section…
I would recommend to the authors to pay more attention to the structure of the reporting data from every guideline like they did in the table.
From the manuscript and from the table relationships between DEX and PRP are not clear. In the table laser mostly referred to the macular procedures, while in the text PRP is discussed only in a single paragraph (354).
66 not clear which parameter of ORL authors mean
123 sounds like DEX restores retinal tissue which is definitely not possible
170 what about brolucizumab?
201 RNDP – please spell out
258 fc – please spell out
275 please make sure that DEX implants can be freely used in pregnancy and breastfeeding without any precautions.
307-308 this phrase seems to belong to etiopathogenesis section
306 I fail to find in this section what in fact is the first line treatment
354 please spell out PRP. Please check whole manuscript for abbreviations
373-374 percentage of what? Diabetic population?
383-394 I think this part is not necessary
429 please clarify this for different types of anomalies of vitreoretinal interface (probably extrapolating from anti-VEGF experience). Please also provide similar data from other guidelines if any
434-502 I think this part could be shorten and embedded into recommendations of Italian guidelines section
Table. Should you include topical steroid test in the table if this is not considered in any guidelines?
519 DRIL not DRILL
520 sounds strange, like we do not know when initiate DME treatment, please check.
535 topical steroids test unexpectedly appears in several point with the manuscript, without any rationale or explanation
Author Response
Dr Spinetta et al provides a review of several national recommendations on the management of DME focusing on the use of biodegradable intraocular corticosteroid implants, mostly dexamethasone. Although, this analysis sounds interesting, I find data from different sources not equal at least in authors’ representation. It is not clear for me why Italian guidelines for cataract are put in a separate section…
I would recommend to the authors to pay more attention to the structure of the reporting data from every guideline like they did in the table.
We thank the reviewer for the comment. We have modified the text as requested.
We have included the guidelines for the use of dexamethasone in patients undergoing cataract surgery in a separate section as it was a different paper from the Italian national guidelines and to underline that this paper only took this aspect into consideration. We considered important to add this paragraph given the importance of the comorbidity between cataracts and diabetes and to stimulate further future work in this directions.
- A: From the manuscript and from the table relationships between DEX and PRP are not clear. In the table laser mostly referred to the macular procedures, while in the text PRP is discussed only in a single paragraph (354).
R: We thanks the reviewer for the comment. We decided to cancel the table of laser because it was outside the scope of our review.
- A: 66 not clear which parameter of ORL authors mean
R: Thanks for the comment, but we couldn’t find the term ORL in that line.
- A: 123 sounds like DEX restores retinal tissue which is definitely not possible
R: We thank the reviewer for the comment. We agree and we have explained it better in the text.
- A: 170 what about brolucizumab?
R: We thank the reviewer for the comment. We follow the suggestion and we added brolucizumab and faricimab.
- A: 201 RNDP – please spell out
R: We thank the reviewer for the comment. We have modified as requested.
- A: 258 fc – please spell out
R: We thank the reviewer for the comment. We have modified as requested
- A: 275 please make sure that DEX implants can be freely used in pregnancy and breastfeeding without any precautions.
R: We thank the reviewer for the comment. We agree that the DEX implant should be use carefully in those situations, so we have explained better the possible use of DEX-I in pregnancy and breastfeeding
- A: 307-308 this phrase seems to belong to etiopathogenesis section
R: We thank the reviewer for the comment. We totally agree so we delete this phrase
- A: 306 I fail to find in this section what in fact is the first line treatment
R: both antiVEGFs and Dex implant are considered first line treatments as written few lines below "the experts agreed that Dex implant is a valid alternative in first line therapy of DME ,similar in efficacy to antiVEGF therapy, in particular for pseudophakic patients and that it should be the first choice therapy in vitrectomized patients
- A: 354 please spell out PRP. Please check whole manuscript for abbreviations
R: We thank the reviewer for the comment. We have checked all abbreviations.
- 373-374 percentage of what? Diabetic population?
R: We thank the reviewer for the comment. We explained it better in the text.
- A: 383-394 I think this part is not necessary
R: We agree with the reviewer and we modified the text as requested
- A: 429 please clarify this for different types of anomalies of vitreoretinal interface (probably extrapolating from anti-VEGF experience). Please also provide similar data from other guidelines if any
R: We thank the reviewer for the comment. We rewrote the sentence more clearly.
- A: 434-502 I think this part could be shorten and embedded into recommendations of Italian guidelines section
R: We thank the reviewer for the comment. We have included the guidelines for the use of dexamethasone in patients undergoing cataract surgery in a separate section as it was a different paper from the Italian national guidelines and to underline that this paper only took this aspect into consideration. We considered important to add this paragraph given the importance of the comorbidity between cataracts and diabetes and to stimulate further future work in this directions.
- A: Table. Should you include topical steroid test in the table if this is not considered in any guidelines?
R: We thank the reviewer for the comment. We included because it was considered in Italian guidelines.
- A: 519 DRIL not DRILL
R: We thank the reviewer for the comment. We have modified the text.
- A: 520 sounds strange, like we do not know when initiate DME treatment, please check.
R: We thank the reviewer for the comment. We have modified the text explaining it better.
- A: 535 topical steroids test unexpectedly appears in several point with the manuscript, without any rationale or explanation
R: We thank the reviewer for the comment. It is known in literature that the use of topical steroids before cortisone implantation has been used, with often discordant results according to publications, as a predictor of the rise in intraocular pressure. In the various consensuses that we have analyzed, only the Italian guidelines (as specified in the table) took it into consideration. So, for completeness we decided to include it. We have tried to clarify the concept in the new version of discussion.
Reviewer 3 Report
The author summarized and compared expert consensus on Intravitreal DEX implant treatment of DME, which has certain reference significance for clinical medication.
Suggestion: The discussion section should be categorized according to the content in Table 2 and Table 3, with subheadings and more detailed content.
Quality of English Language is fine.
Author Response
The author summarized and compared expert consensus on Intravitreal DEX implant treatment of DME, which has certain reference significance for clinical medication.
Suggestion: The discussion section should be categorized according to the content in Table 2 and Table 3, with subheadings and more detailed content.
We thank the reviewer for the comments. We modified the discussion following the suggestions.
Reviewer 4 Report
The authors summarized several countries' diverse suggestions to develop evidence-based guidance on using intravitreal DEX in treating DME. I believe that this kind of effort is necessary to increase efficiency and accuracy in the treatment of DME. The review is well-organized and addresses the current issues in this field. So, I support this review to be published in Pharmaceutics without modification.
Author Response
The authors summarized several countries' diverse suggestions to develop evidence-based guidance on using intravitreal DEX in treating DME. I believe that this kind of effort is necessary to increase efficiency and accuracy in the treatment of DME. The review is well-organized and addresses the current issues in this field. So, I support this review to be published in Pharmaceutics without modification.
We thank the referee for the comments.
Round 2
Reviewer 2 Report
Many thanks to the authors. All questions were adequately answered.